# HIERARCHY-GUIDED MULTIMODAL REPRESENTATION LEARNING FOR TAXONOMIC INFERENCE

**Sk Miraj Ahmed**[1], **Xi Yu**[1], **Yunqi Li**[2]*, **Yuewei Lin**[1], **Wei Xu**[1]

[1]Computing and Data Sciences, Brookhaven National Laboratory, Upton, NY 11973, USA
[2]Rutgers University, New Brunswick, NJ, USA
[1]{sahmed3, xyu1, ywlin, xuw}@bnl.gov
[2]yunqi.li@rutgers.edu

## ABSTRACT

Accurate biodiversity identification from large-scale field data is a foundational problem with direct impact on ecology, conservation, and environmental monitoring. In practice, the core task is *taxonomic prediction*—inferring order, family, genus, or species from imperfect inputs such as specimen images, DNA barcodes, or both. Existing multimodal methods often treat taxonomy as a flat label space and therefore fail to encode the hierarchical structure of biological classification, which is critical for robustness under noise and missing modalities. We present two end-to-end variants for hierarchy-aware multimodal learning: **CLiBD-HiR**, which introduces *Hierarchical Information Regularization (HiR)* to shape embedding geometry across taxonomic levels, yielding structured and noise-robust representations; and **CLiBD-HiR-Fuse**, which additionally trains a *lightweight fusion predictor* that supports image-only, DNA-only, or joint inference and is resilient to modality corruption. Across large-scale biodiversity benchmarks, our approach improves taxonomic classification accuracy by over 14% compared to strong multimodal baselines, with particularly large gains under partial and corrupted DNA conditions. These results highlight that explicitly encoding biological hierarchy, together with flexible fusion, is key for practical biodiversity foundation models. ⌗ https://github.com/mirajucr/CLIBD-HIR-FUSION

## 1 INTRODUCTION

Biodiversity research Van Horn et al. (2018) increasingly relies on large-scale multimodal data, including specimen images, DNA barcodes, and auxiliary taxonomic metadata. While curated datasets enable controlled benchmarking, real-world deployments are far less controlled. In particular, DNA barcodes obtained through large-scale sequencing pipelines (e.g., BOLD Ratnasingham & Hebert (2007)) can exhibit partial reads, ambiguous bases, and sequencing artifacts Médigue et al. (1999); Meiklejohn et al. (2019), and field-collected specimen images are often degraded by cluttered backgrounds, occlusions, lighting variation, motion blur Nguyen et al. (2024), or low signal-to-noise acquisition. Bridging this gap between curated evaluation and imperfect operational inputs makes robust taxonomic inference from heterogeneous and noisy modalities a central and unresolved challenge in applied biodiversity science.

A key recent step toward unifying these modalities is **CLIBD** Gong et al. (2024), which to our knowledge is the only prior work that explicitly aligns *images, DNA barcodes, and taxonomic text* in a shared embedding space at *very large scale*, and is therefore our primary state-of-the-art reference. In practice, however, cross-modal retrieval typically surfaces candidate matches that still require downstream verification by human experts, whereas operational pipelines ultimately benefit from reliable *taxonomic prediction* from one or more available modalities. More importantly, existing multimodal objectives commonly treat taxonomy as a flat label space and rely on standard contrastive learning without explicitly encoding biological hierarchy. As a result, learned embeddings may lack hierarchy-consistent geometry: closely related taxa are not guaranteed to be nearby, and perturbations from noise or missing data can lead to unpredictable errors across taxonomic levels. This issue is particularly acute when one modality—most commonly DNA—is partially

---

*This author contributed to this work while at Brookhaven National Laboratory.

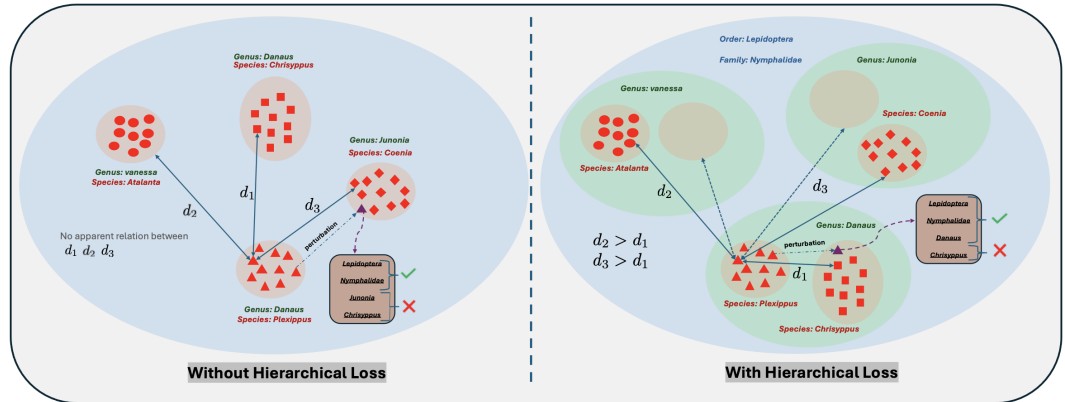

Figure 1: **Effect of hierarchical regularization on embedding geometry and noise robustness (CLIBD-HiR, variant 1). Left:** without the HiR loss, standard contrastive training treats mismatched taxa uniformly, yielding no explicit geometric relationship between intra-genus distances ($d_1$; different species within the same genus) and inter-genus / higher-level distances ($d_2, d_3$). Under realistic noise, a perturbed query embedding may drift across arbitrary clusters, leading to errors that can propagate to higher taxonomic ranks. **Right:** with HiR, the loss explicitly enforces a hierarchy-consistent structure ($d_1 < d_2 < d_3$), so nearby neighborhoods reflect taxonomic proximity. Consequently, even when noise causes a species-level mistake, predictions are more likely to remain correct at coarser levels (genus/family/order), improving robustness.

corrupted or unavailable, a routine scenario in large-scale biodiversity repositories. Finally, most existing approaches do not explicitly model *adaptive image–DNA fusion*, even though DNA alone may be insufficient in practice and complementary morphological cues from images can be critical for resolving fine-grained taxa Cong et al. (2017).

In this work, we build on CLIBD Gong et al. (2024) and present a taxonomy-aware multimodal framework for robust taxonomic prediction from images, DNA barcodes, and their combination under realistic noise. Our approach is instantiated as two complementary end-to-end variants of CLIBD (Algo 1 and 2) that (i) inject taxonomic hierarchy into representation learning via a hierarchy-aware objective, and (ii) optionally train an explicit image–DNA fusion predictor for joint inference.

**Algo 1: CLIBD-HiR (structured, noise-robust representation learning).** Our first variant targets a core failure mode of prior contrastive methods: the lack of hierarchy-consistent structure in the learned representation space. We propose **Hierarchical Information Regularization (HiR)**, which injects taxonomic hierarchy directly into representation learning and explicitly *shapes the embedding geometry* during training. Samples sharing coarser taxa (e.g., family or genus) are encouraged to remain close, while finer distinctions (e.g., species) are learned without collapsing higher-level neighborhoods. This hierarchical organization acts as an intrinsic noise stabilizer: when a noisy sample drifts away from its species cluster due to image corruption or barcode degradation, HiR still anchors it to the correct higher-level neighborhood, limiting catastrophic semantic drift (Fig. 1).

**Algo 2: CLIBD-HiR-Fuse (adaptive fusion for variable modality quality).** Our second variant extends CLIBD-HiR with a lightweight fusion predictor trained jointly with the encoders. This is motivated by deployment realities where available evidence varies by sample—some specimens have only images, some only barcodes, and some both, often with differing degrees of corruption. CLIBD-HiR-Fuse (Fig. 2) supports image-only, DNA-only, and fused image+DNA inference, and leverages the hierarchy-aware aligned space to better combine complementary signals when one modality is unreliable.

Together, hierarchy-guided regularization (Algo 1) and adaptive fusion (Algo 2) produce noise-resilient multimodal models that better match real-world biodiversity workflows. Across large-scale benchmarks, our approach improves taxonomic prediction accuracy over CLIBD and strong fusion baselines, with particularly large gains under DNA corruption and low-quality imaging.

**Our contributions are threefold:**

• We introduce **Hierarchical Information Regularization (HiR)**, a taxonomy-aware objective that explicitly shapes embedding geometry and improves robustness to noisy and partially corrupted inputs.
• We present two end-to-end variants: **CLIBD-HiR** (Algo 1), a structured embedding learner optimized for hierarchical taxonomic prediction, and **CLIBD-HiR-Fuse** (Algo 2), which adds an adaptive fusion predictor supporting image-only, DNA-only, and image+DNA inference under varying modality quality.
• We demonstrate consistent improvements in taxonomic prediction across biodiversity benchmarks, with especially large gains in noise-dominated regimes.

## 2 RELATED WORK

**Foundation models for biodiversity and multimodal alignment.** Recent progress in foundation models has enabled transferable representations via large-scale pretraining and multimodal alignment. CLIP-style training aligns vision and language to support zero-shot transfer (Radford et al., 2021; Cherti et al., 2023), and newer multimodal alignment frameworks extend this idea beyond image–text to jointly embed heterogeneous modalities. For example, ImageBind learns a shared space across multiple sensory streams (e.g., image, text, audio, depth, IMU) using paired data (Girdhar et al., 2023), and related efforts such as "X-CLIP"/"LanguageBind"-style models extend CLIP-like alignment to additional modalities. In biodiversity, BioCLIP adapts CLIP-style pretraining toward organism-centric visual recognition (Stevens et al., 2024), while CLIBD is a key step toward *multimodal* biodiversity foundation modeling by aligning specimen images, DNA barcodes, and taxonomic text in a shared embedding space (Gong et al., 2024; Gharaee et al., 2023). These backbones provide strong representations, but are typically optimized for retrieval-style alignment and do not explicitly enforce hierarchy-consistent geometry or robustness to modality degradation.

**Taxonomy-aware representation learning.** Taxonomic labels are inherently hierarchical (order–family–genus–species), motivating objectives that respect coarse-to-fine structure beyond flat classification. Prior work has explored hierarchy-aware losses and hierarchical contrastive learning to impose semantic structure directly in the embedding space (Khosla et al., 2020; Zhang et al., 2022). Our HiR regularization builds on this line by injecting taxonomic hierarchy into multimodal alignment, shaping neighborhoods so that nearby embeddings reflect biological relatedness and improving robustness when fine-grained cues are noisy or incomplete.

**Multimodal fusion under noisy modalities.** Beyond alignment, practical biodiversity applications often benefit from combining complementary evidence across modalities. Recent fusion strategies include uncertainty-aware fusion that explicitly improves robustness to noisy unimodal representations (Gao et al., 2024), as well as dynamic routing / mixture-of-experts style fusion with learned gating that adapts fusion behavior to the input (Cao et al., 2023; Han et al., 2024). We build on these ideas with a lightweight gated fusion head that adaptively mixes image and DNA embeddings, and evaluate it against naive averaging under clean and degraded modality conditions.

**Robust modeling of barcodes and field imagery.** DNA barcodes encountered in operational pipelines can be imperfect (e.g., substitutions/indels, ambiguous bases, partial reads), motivating noise-aware preprocessing and modeling Médigue et al. (1999); similarly, field imagery is affected by background clutter, occlusion, illumination/pose changes, and motion blur, which can degrade fine-grained recognition Nguyen et al. (2024). Our work is complementary: rather than relying on noise-specific training data, we stabilize multimodal prediction by enforcing hierarchy-consistent geometry and enabling adaptive fusion to leverage complementary evidence when one modality is degraded.

## 3 METHOD

We consider triplets $(x_i^V, x_i^D, x_i^T)$ consisting of an image, a DNA barcode, and a textual taxonomy description for specimen $i$. Our goal is twofold: (1) learn a shared multimodal embedding space

where visual, DNA, and text representations are aligned in a hierarchy-aware, noise-robust manner; and (2) learn a fusion network that directly predicts taxonomy from jointly using image and DNA features.

We denote the encoders by

$$\mathbf{v}_i = f_V(x_i^V), \quad \mathbf{d}_i = f_D(x_i^D), \quad \mathbf{t}_i = f_T(x_i^T),$$

where $\mathbf{v}_i, \mathbf{d}_i, \mathbf{t}_i \in \mathbb{R}^d$ are $\ell_2$-normalized embeddings. Each specimen is also annotated with hierarchical taxonomic labels

$$\mathbf{y}_i = \left(y_i^{(1)}, y_i^{(2)}, \ldots, y_i^{(L)}\right),$$

e.g., order, family, genus, species for $L = 4$.

Our framework is trained in two stages: (1) multimodal pretraining with symmetric cross-modal contrastive losses plus an image-only Hierarchical Information Regularization (HiR) loss; and (2) post-hoc training of an MLP fusion head that predicts taxonomy from visual and DNA embeddings.

## 3.1 Cross-Modal Contrastive Objectives

Following CLIBD Gong et al. (2024), we align modalities with symmetric InfoNCE. For a batch of size $N$, define $s(\mathbf{a}_i, \mathbf{b}_j) = \mathbf{a}_i^\top \mathbf{b}_j / \tau$. The directed loss is

$$\mathcal{L}_{A \to B} = -\frac{1}{N} \sum_{i=1}^{N} \log \frac{\exp(s(\mathbf{a}_i, \mathbf{b}_i))}{\sum_{j=1}^{N} \exp(s(\mathbf{a}_i, \mathbf{b}_j))}, \tag{1}$$

and $\mathcal{L}_{A \leftrightarrow B} = \mathcal{L}_{A \to B} + \mathcal{L}_{B \to A}$. We use pairs $(V, T)$, $(V, D)$, and $(D, T)$, and combine them as

$$\mathcal{L}_{\text{XMOD}} = \lambda_{VT} \mathcal{L}_{V \leftrightarrow T} + \lambda_{VD} \mathcal{L}_{V \leftrightarrow D} + \lambda_{DT} \mathcal{L}_{D \leftrightarrow T}. \tag{2}$$

## 3.2 Hierarchical Information Regularization for Images

While the cross-modal objective encourages alignment between modalities, it does not explicitly encode the taxonomy hierarchy, nor does it guarantee robustness when one modality is noisy or partially corrupted. To address this, we introduce an image-only hierarchical contrastive loss inspired by the HiConE objective from Zhang et al. (2022).

For each taxonomic level $\ell \in \{1, \ldots, L\}$ (e.g., order, family, genus, species), we treat images sharing the same label $y_i^{(\ell)}$ as positives at level $\ell$. Let $\mathcal{P}_i^{(\ell)}$ be the set of indices $j \neq i$ such that $y_j^{(\ell)} = y_i^{(\ell)}$, and let $\mathcal{N}_i^{(\ell)}$ be the remaining images in the batch. For an anchor $i$ and a positive $j \in \mathcal{P}_i^{(\ell)}$, we define the *pair-wise* supervised contrastive loss at level $\ell$ as

$$\ell^{(\ell)}(i, j) = -\log \frac{\exp\left(s(\mathbf{v}_i, \mathbf{v}_j)\right)}{\sum_{k \in \mathcal{P}_i^{(\ell)} \cup \mathcal{N}_i^{(\ell)}} \exp\left(s(\mathbf{v}_i, \mathbf{v}_k)\right)}. \tag{3}$$

The standard level-$\ell$ supervised contrastive loss is then

$$\mathcal{L}_{\text{SupCon}}^{(\ell)} = \frac{1}{N} \sum_{i=1}^{N} \frac{1}{|\mathcal{P}_i^{(\ell)}|} \sum_{j \in \mathcal{P}_i^{(\ell)}} \ell^{(\ell)}(i, j). \tag{4}$$

Following HiConE, we further enforce that *finer* taxonomic levels cannot be optimized in a way that violates coarser-level structure. To this end, we compute, for each level $\ell$, the maximum pair-wise loss over all positive pairs at that level:

$$m^{(\ell)} = \max_i \max_{j \in \mathcal{P}_i^{(\ell)}} \ell^{(\ell)}(i, j), \tag{5}$$

and define a *hierarchically rectified* pair loss

$$\tilde{\ell}^{(1)}(i, j) = \ell^{(1)}(i, j), \quad \tilde{\ell}^{(\ell)}(i, j) = \max\left(\ell^{(\ell)}(i, j), \, m^{(\ell-1)}\right) \text{ for } \ell > 1. \tag{6}$$

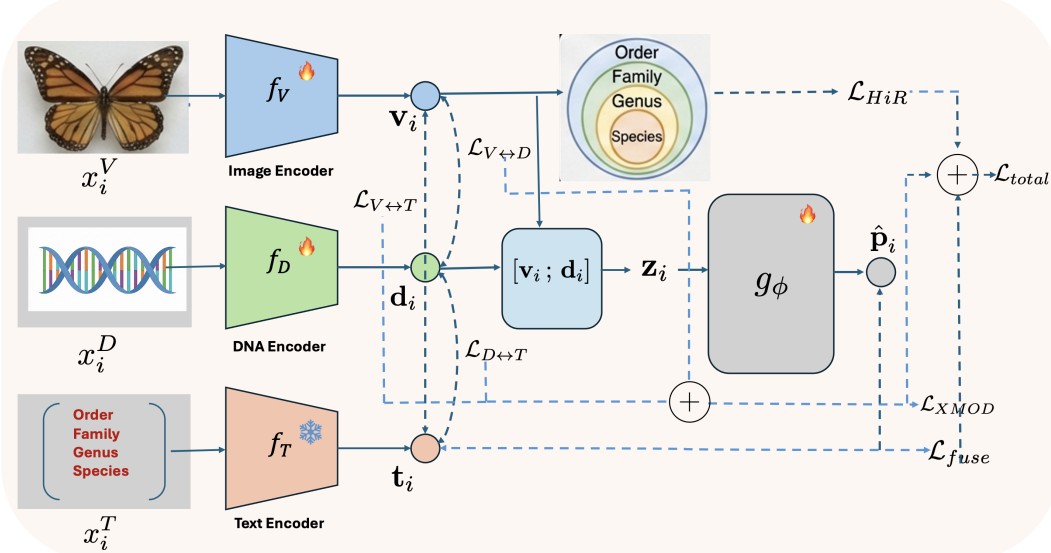

Figure 2: **CLIBD-HiR-Fuse framework (Algorithm variant 2).** Given a specimen image and its DNA barcode, we encode each modality with an image encoder and a DNA encoder, and embed the taxonomy prompt with a frozen BioCLIP text encoder. We align image–text and DNA–text representations using CLIP-style contrastive learning, and enforce hierarchy-aware structure with a hierarchical loss over augmented image views. A lightweight GatedFusion module adaptively combines image and DNA embeddings into a fused representation, which is additionally aligned to the fixed text embedding space via a fused-to-text contrastive objective.

Intuitively, this loss prevents the model from minimizing fine-grained losses while coarser-level structure is still poorly organized. Concretely, if the loss of a fine-level positive pair (e.g., same species) becomes *smaller* than the largest loss among positives at the immediately coarser level (e.g., same genus), we *clamp* the fine-level loss to that coarser-level maximum. This ensures that fine-level positives are not allowed to be optimized "ahead" of the coarser level: the model must first reduce the worst-case within-genus (or within-family) positive loss before further tightening species-level clusters. Our final Hierarchical Information Regularization (HiR) loss aggregates these rectified pair losses across levels:

$$\mathcal{L}_{\text{HiR}} = \sum_{\ell=1}^{L} \alpha_\ell \left[ \frac{1}{N} \sum_{i=1}^{N} \frac{1}{|\mathcal{P}_i^{(\ell)}|} \sum_{j \in \mathcal{P}_i^{(\ell)}} \tilde{\ell}^{(\ell)}(i,j) \right], \tag{7}$$

where $\alpha_\ell$ are non-negative weights (we use uniform weights in our experiments).

This hierarchical structure makes the visual encoder *noise-robust*: if an embedding is perturbed such that it drifts away from its species cluster, the loss still anchors it using genus and family supervision, and the max-rectification in Eq. 6 prevents the optimizer from overfitting to noisy fine-grained labels while ignoring coarser, more reliable signals. As a result, even under noisy supervision or mismatched modalities, the image representation preserves higher-level semantic consistency (Fig. 1).

### 3.3 OBJECTIVES: TWO END-TO-END VARIANTS (ALGO 1 VS. ALGO 2)

We instantiate our framework in two variants that share the same multimodal alignment and hierarchical regularization backbone, but differ in whether an explicit fusion predictor is trained.

**Algo 1: CLiBD-HiR (structured, noise-robust representation learning).** In Algo 1, we train the encoders end-to-end using the sum of cross-modal contrastive alignment and hierarchical image regularization:

$$\mathcal{L}_{\text{total}}^{(1)} = \mathcal{L}_{\text{XMOD}} + \lambda_{\text{HiR}} \mathcal{L}_{\text{HiR}}, \tag{8}$$

where $\lambda_{\text{HiR}}$ balances hierarchical image regularization against multimodal alignment. This objective encourages a hierarchy-consistent embedding geometry, improving robustness when one modality is noisy or partially corrupted, and yielding strong taxonomy classification via nearest-neighbor or linear probing in the learned space. (Full pseudocde in Algorithm 1).

**Algo 2: CLiBD-HiR-Fuse (flexible fusion with missing-modality support).** Algo 2 augments Algo 1 with a lightweight fusion module trained *jointly* with the encoders under a single objective:

$$\mathcal{L}_{\text{total}}^{(2)} = \mathcal{L}_{\text{XMOD}} + \lambda_{\text{HiR}}\mathcal{L}_{\text{HiR}} + \lambda_{\text{fuse}}\mathcal{L}_{\text{fuse}}, \tag{9}$$

where $\lambda_{\text{fuse}}$ controls the contribution of fusion loss $\mathcal{L}_{\text{fuse}}$ (Eq. 11). This variant provides a direct prediction interface that is applicable when only one modality is available (image-only or DNA-only) and when both modalities are available, while remaining noise-resilient due to the shared aligned and hierarchy-aware representation. (Full pseudocde in Algorithm 2)

## 3.4 FLEXIBLE MULTIMODAL FUSION FOR TAXONOMY PREDICTION (ALGO 2)

Algo 2 introduces a lightweight fusion predictor $g_\phi$ that maps available non-text modality embeddings to taxonomy logits. Given image and DNA embeddings, we concatenate them as $\mathbf{z}_i = [\mathbf{v}_i; \mathbf{d}_i] \in \mathbb{R}^{2d}$ and define the fused representation as

$$\hat{\mathbf{p}}_i = \mathbf{v}_i \text{ (image-only)}, \quad \mathbf{d}_i \text{ (DNA-only)}, \quad g_\phi(\mathbf{z}_i) \text{ (image+DNA)}. \tag{10}$$

We train $g_\phi$ with a supervised cross-entropy loss at level $\ell^\star$,

$$\mathcal{L}_{\text{fuse}} = -\frac{1}{N}\sum_{i=1}^{N} \log \hat{\mathbf{p}}_i\big[y_i^{(\ell^\star)}\big]. \tag{11}$$

Since the encoders are aligned across modalities and regularized to respect taxonomy, the fusion predictor is more robust when one modality is degraded.

**Degradation Models.** To assess robustness under realistic perturbations, we introduce degradation strategies for both modalities. For images, we simulate optical blur and defocus by convolving the input $x$ with a normalized averaging kernel $h \in \mathbb{R}^{k \times k}$, where the kernel size $k$ controls the severity of high-frequency attenuation. For DNA, we model sequencing errors and partial reads by transforming a clean sequence $s$ into a corrupted observation $\tilde{s}$ via a stochastic pipeline comprising five operations: (1) *substitution* of nucleotides with probability $p_{\text{sub}}$; (2) *ambiguous masking* where bases are replaced by 'N' with probability $p_{\text{mask}}$; (3) *insertions and deletions* ($p_{\text{ins}}, p_{\text{del}}$) to simulate frameshifts; (4) *contiguous dropout* of a subsequence with relative length $\rho$; and (5) *tail truncation* ($\tau$) to mimic incomplete reads.

## 4 EXPERIMENTS

**Dataset and split.** We use the BIOSCAN-1M Gharaee et al. (2023) insect dataset and construct paired samples consisting of a specimen image, a COI DNA barcode sequence, and a textual taxonomy description derived from the hierarchical labels (order, family, genus, species). Each sample includes both string taxonomy fields and consistent integer label IDs at each taxonomic level. Our final split contains 903,536 training samples and 224,777 test samples. The split is *closed-set* across all taxonomic levels: every order, family, genus, and species that appears in the test set also appears in the training set (no unseen taxa in test). Taxonomic completeness varies across specimens (many are labeled only up to coarser levels such as order or family); accordingly, we report evaluation at each level using all available labels for that level.

### 4.1 BASELINES AND EVALUATION METRICS

We use CLIBD Gong et al. (2024) as the primary reference and construct all comparisons to isolate the effects of hierarchical regularization and fusion. **(1) No-fusion comparison (Table 1).** We report **CLIBD** and **CLIBD-HiR** under clean and noisy settings, evaluating Image→Text and DNA→Text Top-1/Top-5 taxonomic classification accuracy without any Image–DNA fusion. **(2) Fusion comparison within CLIBD-HiR (Table 2).** Here we keep the same CLIBD-HiR training setup and

---

**Algorithm 1** CLiBD-HiR: End-to-End Hierarchy-Guided Multimodal Contrastive Training

---

**Require:** Encoders $f_V, f_D, f_T$; temperature $\tau$; weights $\lambda_{VT}, \lambda_{VD}, \lambda_{DT}, \lambda_{\text{HiR}}$; hierarchy weights $\{\alpha_\ell\}_{\ell=1}^L$
1: **while** not converged **do**
2:     Sample mini-batch $\{(x_i^V, x_i^D, x_i^T, \mathbf{y}_i)\}_{i=1}^N$
3:     **for** $i = 1$ to $N$ **do**
4:         $\mathbf{v}_i \leftarrow f_V(x_i^V), \ \mathbf{d}_i \leftarrow f_D(x_i^D), \ \mathbf{t}_i \leftarrow f_T(x_i^T)$
5:         (Optional) $\ell_2$-normalize $\mathbf{v}_i, \mathbf{d}_i, \mathbf{t}_i$
6:     **end for**
7:     Compute pairwise similarities with temperature $\tau$
8:     Compute symmetric CLIP-style losses $\mathcal{L}_{V \leftrightarrow T}, \mathcal{L}_{V \leftrightarrow D}, \mathcal{L}_{D \leftrightarrow T}$
9:     $\mathcal{L}_{\text{CLIBD}} \leftarrow \lambda_{VT} \mathcal{L}_{V \leftrightarrow T} + \lambda_{VD} \mathcal{L}_{V \leftrightarrow D} + \lambda_{DT} \mathcal{L}_{D \leftrightarrow T}$
10:     $\mathcal{L}_{\text{HiR}} \leftarrow 0, \ m^{(0)} \leftarrow 0$
11:     **for** $\ell = 1$ to $L$ **do**             ▷ taxonomy levels: order, family, genus, species
12:         **for** $i = 1$ to $N$ **do**
13:             $\mathcal{P}_i^{(\ell)} \leftarrow \{j \neq i \mid y_j^{(\ell)} = y_i^{(\ell)}\}, \ \mathcal{N}_i^{(\ell)} \leftarrow \{k \neq i \mid y_k^{(\ell)} \neq y_i^{(\ell)}\}$
14:         **end for**
15:         Compute $\ell^{(\ell)}(i, j)$ using Eq. 3
16:         $m^{(\ell)} \leftarrow \max_{i,j \in \mathcal{P}_i^{(\ell)}} \ell^{(\ell)}(i, j)$
17:         **if** $\ell = 1$ **then**
18:             $\tilde{\ell}^{(\ell)}(i, j) \leftarrow \ell^{(\ell)}(i, j)$
19:         **else**
20:             $\tilde{\ell}^{(\ell)}(i, j) \leftarrow \max\big(\ell^{(\ell)}(i, j), \ m^{(\ell-1)}\big)$
21:         **end if**
22:         $\mathcal{L}_{\text{HiR}}^{(\ell)} \leftarrow \frac{1}{N} \sum_{i=1}^N \frac{1}{|\mathcal{P}_i^{(\ell)}|} \sum_{j \in \mathcal{P}_i^{(\ell)}} \tilde{\ell}^{(\ell)}(i, j)$
23:         $\mathcal{L}_{\text{HiR}} \leftarrow \mathcal{L}_{\text{HiR}} + \alpha_\ell \mathcal{L}_{\text{HiR}}^{(\ell)}$
24:     **end for**
25:     $\mathcal{L}_{\text{total}} \leftarrow \mathcal{L}_{\text{CLIBD}} + \lambda_{\text{HiR}} \mathcal{L}_{\text{HiR}}$
26:     Update parameters of $f_V, f_D, f_T$ using $\nabla \mathcal{L}_{\text{total}}$
27: **end while**
28: **return** Trained encoders $f_V, f_D, f_T$

---

compare three variants: **CLIBD-HiR** (no fusion; unimodal I→T and D→T), **CLIBD-HiR + Avg** (naïve averaging of image and DNA embeddings for I+D→T), and **CLIBD-HiR + Fusion** (our learned GatedFusion head for I+D→T). We evaluate robustness under both DNA-only noise (*Noisy D*) and joint image+DNA noise (*Noisy I+D*). This isolates the benefit of adaptive fusion beyond hierarchy-aware representation learning. As per the metrics, we report Top-1 and Top-5 taxonomic prediction accuracy at four hierarchical levels (order, family, genus, species). For each query (image, DNA, or fused image+DNA), we rank candidate taxonomy text prompts by similarity in the shared embedding space and measure whether the ground-truth label appears at rank 1 or within the top 5. **Global** denotes an aggregate accuracy across the four taxonomic levels.

**Models and training setup.** We adopt a three-encoder architecture consisting of an image encoder, a DNA encoder, and a text encoder. The image and text encoders are initialized from a pretrained vision–language model, either standard OpenCLIP ViT-L/14 (Cherti et al., 2023) or Bio-CLIP ViT-L/14 (Stevens et al., 2024). The DNA branch is built upon DNABERT2 (Zhou et al., 2024). When BioCLIP is used, we employ a modified DNABERT2 variant by adding a learnable linear projection layer on top of the DNABERT2 embedding to match the embedding dimension of the image–text backbone. Following Algo. 1 and Algo. 2, we train the image and DNA encoders end-to-end. The optimization strategy for the text encoder depends on the chosen backbone. With BioCLIP, we freeze the entire text encoder. With standard OpenCLIP, we fine-tune the text encoder during training. Empirically, freezing BioCLIP text encoder improves performance, likely because BioCLIP already encodes strong biological language priors (Stevens et al., 2024); this differs from the original CLIBD training recipe, which fine-tunes the text encoder (Gong et al., 2024). Unless stated otherwise, we report results using this *fixed-BioCLIP-text* variant. Training uses contrastive alignment losses together with a hierarchy-aware loss on augmented image features (Khosla

---

**Algorithm 2** CLiBD-HiR-Fuse: End-to-End Multimodal Contrastive + HiR + Fusion-Supervised Training

---

**Require:** Encoders $f_V, f_D, f_T$; fusion head $g_\phi$; temperature $\tau$; weights $\lambda_{VT}, \lambda_{VD}, \lambda_{DT}, \lambda_{\text{HiR}}, \lambda_{\text{fuse}}$; hierarchy weights $\{\alpha_\ell\}_{\ell=1}^L$

1: **while** not converged **do**
2:     Sample mini-batch $\{(x_i^V, x_i^D, x_i^T, \mathbf{y}_i)\}_{i=1}^N$
3:     **for** $i = 1$ to $N$ **do**
4:         $\mathbf{v}_i \leftarrow f_V(x_i^V)$, $\mathbf{d}_i \leftarrow f_D(x_i^D)$, $\mathbf{t}_i \leftarrow f_T(x_i^T)$
5:         (Optional) $\ell_2$-normalize $\mathbf{v}_i, \mathbf{d}_i, \mathbf{t}_i$
6:         $\mathbf{z}_i \leftarrow [\mathbf{v}_i \,;\, \mathbf{d}_i]$                           ▷ fusion input
7:         $\hat{\mathbf{p}}_i \leftarrow g_\phi(\mathbf{z}_i)$                  ▷ taxonomy logits/probabilities
8:     **end for**
9:     Compute similarities with temperature $\tau$
10:    Compute symmetric CLIP-style losses $\mathcal{L}_{V \leftrightarrow T}, \mathcal{L}_{V \leftrightarrow D}, \mathcal{L}_{D \leftrightarrow T}$
11:    $\mathcal{L}_{\text{CLIBD}} \leftarrow \lambda_{VT}\mathcal{L}_{V \leftrightarrow T} + \lambda_{VD}\mathcal{L}_{V \leftrightarrow D} + \lambda_{DT}\mathcal{L}_{D \leftrightarrow T}$
12:    Compute hierarchy-guided loss $\mathcal{L}_{\text{HiR}}$ exactly as in Alg. 1
13:    Compute fusion classification loss $\mathcal{L}_{\text{fuse}}$ (Eq. 11)
14:    $\mathcal{L}_{\text{total}} \leftarrow \mathcal{L}_{\text{CLIBD}} + \lambda_{\text{HiR}}\mathcal{L}_{\text{HiR}} + \lambda_{\text{fuse}}\mathcal{L}_{\text{fuse}}$
15:    Update parameters of $f_V, f_D, f_T, g_\phi$ using $\nabla\mathcal{L}_{\text{total}}$
16: **end while**
17: **return** Trained encoders $f_V, f_D, f_T$ and fusion head $g_\phi$

---

et al., 2020; Zhang et al., 2022). For the fusion variant, we add a trainable *GatedFusion* head, a lightweight 2-layer MLP (Linear–ReLU–Dropout–Linear–Sigmoid) that predicts a per-dimension gate from concatenated image and DNA embeddings and mixes them before normalization, supervised by an additional fused-to-text contrastive loss.

**Implementation and noise settings.** We train with distributed data parallel on 4 NVIDIA A100 GPUs for ∼1 day (batch size 30, 10 epochs). Images are loaded from an HDF5 container and paired with DNA barcodes and taxonomy text from CSV metadata. Optimization uses AdamW with a OneCycle schedule (base LR $1\times10^{-6}$, max LR $5\times10^{-5}$), with $\lambda_{\text{HiR}} = 0.99$ and $\alpha = 0.1$ in the gathered contrastive loss; the fusion variant adds a fused-to-text term weighted by $\lambda_{\text{fuse}} = 0.7$ (omitted for the no-fusion baseline). For robustness evaluation, we apply *inference-time* modality degradation only (no noisy data during training): DNA is corrupted with substitutions ($p_{\text{sub}}$=0.01), insertions/deletions ($p_{\text{ins}}$=$p_{\text{del}}$=0.002), masking to N ($p_{\text{mask}}$=0.003), contiguous N-dropout (run fraction 0.05), and tail truncation (10%), while images use blur noise with a 7×7 kernel. We report clean, DNA-noisy, and joint image+DNA noisy results.

**Results.** Table 1 shows that adding HiR to CLIBD improves no-fusion taxonomic prediction, with the largest gains under noise. For I→T, CLIBD-HiR increases Global Top-1 from 75.5 to 78.2 on clean data and from 40.0 to 46.6 on noisy data (Top-5: 58.4 to 59.6), driven mainly by improved coarse-level accuracy (e.g., noisy Family Top-1: 66.8 to 70.1). For D→T, HiR yields a small clean gain (Global Top-1: 94.8 to 95.6) but a substantial robustness improvement under noisy DNA (Global Top-1: 52.4 to 66.0; Global Top-5: 91.5 to 96.9), with a notable increase at the family level (57.1 to 70.3 Top-1). Overall, HiR consistently improves global performance and noise robustness without using fusion. Moreover, Table 2 analyzes fusion within the same CLIBD-HiR training setup. The unimodal baselines (I→T and D→T) establish modality-specific performance, while the third block, **I+D→T (Avg)**, fuses image and DNA by simple embedding averaging (no fusion module). Our learned fusion model, **I+D→T (Ours)**, improves over averaging in the most realistic setting where both modalities are noisy: Global accuracy increases from 85.5/96.5 to 88.0/97.5 (Top-1/Top-5), with the largest gain at species level (54.6/79.9 to 57.4/81.7). Under DNA-only noise, averaging and learned fusion are comparable globally (91.3/97.7 vs. 91.4/98.0), indicating that the main benefit of the fusion module is robustness when image and DNA quality vary simultaneously.

**Limitations.** We do not use the original CLIBD split, which is designed around seen/unseen evaluation across multiple taxonomic levels and is highly imbalanced. In our initial experiments, this split substantially reduces the effective supervision available at deeper levels (genus/species) and makes

Table 1: **Algo 1 (CLIBD-HiR): no-fusion evaluation.** Comparison of CLIBD and CLIBD-HiR without any image–DNA fusion module. We report **Top-1 / Top-5** taxonomic prediction accuracy (%) for Image→Text and DNA→Text under **clean** inputs and under **noisy** inputs (synthetically degraded at inference). **Global** denotes an aggregate across order, family, genus, and species. Highlighted rows indicate our proposed HiR model.

| Setting | Method | Cond. | Top-1 / Top-5 Accuracy | | | | |
|---|---|---|---|---|---|---|---|
| | | | Order | Family | Genus | Species | Global |
| I→T | CLIBD | Clean | 98.8 / 99.2 | 93.3 / 95.8 | **77.8 / 90.4** | **46.6 / 71.4** | 75.5 / **89.8** |
| | | Noisy | **88.8 / 92.1** | 66.8 / 74.7 | **48.7 / 65.6** | **22.2 / 42.0** | 40.0 / 58.4 |
| I→T | **CLIBD-HiR** | Clean | **99.3 / 99.5** | **94.3 / 96.1** | 76.8 / 89.1 | 46.6 / 68.3 | **78.2 / 89.2** |
| | **CLIBD-HiR** | Noisy | 88.7 / 91.4 | **70.1 / 77.0** | 42.9 / 59.0 | 15.5 / 31.8 | **46.6 / 59.6** |
| D→T | CLIBD | Clean | 99.9 / 99.9 | **99.4 / 99.8** | 94.7 / 98.1 | 68.1 / 88.5 | 94.8 / 98.3 |
| | | Noisy | 99.7 / **99.9** | 57.1 / 97.8 | **81.6 / 94.2** | 48.6 / 76.3 | 52.4 / 91.5 |
| D→T | **CLIBD-HiR** | Clean | **100.0 / 100** | 99.3 / **99.9** | **95.9 / 98.8** | **71.6 / 91.3** | **95.6 / 98.7** |
| | **CLIBD-HiR** | Noisy | **99.8 / 99.9** | **70.3 / 99.2** | 81.5 / **95.1** | **51.6 / 79.3** | **66.0 / 96.9** |

Table 2: **Algo 2 (CLIBD-HiR-Fuse): fusion evaluation.** We report **Top-1 / Top-5** taxonomic prediction accuracy (%). Highlighted rows indicate our proposed fusion model. **Bold** indicates the best performance for that specific condition (Clean, Noisy D, or Noisy I+D) across all methods.

| Method | Cond. | Top-1 / Top-5 Accuracy | | | | |
|---|---|---|---|---|---|---|
| | | Order | Family | Genus | Species | Global |
| I→T (Ours) | Clean | 99.3 / 99.5 | 93.5 / 99.9 | 74.7 / 98.9 | 40.4 / 91.8 | 77.0 / 98.7 |
| | Noisy I | 88.3 / 91.8 | 67.0 / 75.9 | 37.0 / 55.7 | 11.7 / 26.7 | 50.2 / 62.1 |
| D→T (Ours) | Clean | **100.0 / 100** | 99.5 / **99.9** | **96.1 / 98.9** | **74.4 / 92.7** | 95.7 / 98.7 |
| | Noisy D | 99.8 / 99.9 | 87.3 / 99.3 | 86.1 / 95.6 | 57.8 / 82.0 | 82.3 / 97.1 |
| I+D→T (Avg) | Clean | 99.9 / **100** | 99.6 / **99.9** | 95.5 / 98.6 | 71.2 / 89.8 | 95.7 / 98.6 |
| | Noisy D | 99.9 / **100** | **96.8 / 99.7** | **90.6 / 97.1** | 60.1 / 82.9 | 91.3 / 97.7 |
| | Noisy I+D | 99.6 / **100** | 91.3 / 99.3 | **87.9 / 96.1** | 54.6 / 79.9 | 85.5 / 96.5 |
| **I+D→T (Ours)** **(GatedFusion)** | Clean | **100.0 / 100** | **99.7 / 99.9** | 96.0 / **98.9** | 73.5 / 91.2 | **96.1 / 98.8** |
| | Noisy D | **100.0 / 100** | 96.2 / **99.7** | 88.7 / 96.9 | **60.8 / 83.2** | **91.4 / 98.0** |
| | Noisy I+D | **99.9 / 100** | **93.0 / 99.5** | 87.0 / 96.0 | **57.4 / 81.7** | **88.0 / 97.5** |

hierarchy learning less informative, leading to performance that is close to the baseline. To clearly demonstrate the impact of hierarchy-aware regularization and fusion, we therefore report results on our split where all taxonomic levels are consistently represented. Developing training and evaluation protocols that better handle the severe long-tail imbalance and level-dependent seen/unseen structure in CLIBD remains an important direction for future work.

## 5 CONCLUSION

We presented HiR-Fusion, a hierarchy-guided multimodal framework for robust taxonomic prediction from images, DNA barcodes, and their combination. Our first variant (CLIBD-HiR) improves noise robustness by explicitly shaping embedding geometry to respect biological hierarchy, and our second variant (CLIBD-HiR-Fuse) further enhances performance by learning an adaptive fusion module that outperforms naive averaging, particularly when both modalities are degraded. Across clean and noisy settings, our results show that incorporating taxonomic structure and reliability-aware fusion yields more robust and practically useful biodiversity recognition models.

ACKNOWLEDGMENTS

This work was supported by the Laboratory Directed Research and Development (LDRD) program through project 24-063.

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
