# OpenReview forum: "Hierarchy-Guided Multimodal Representation Learning for Taxonomic Inference"
_ICLR.cc/2026/Workshop/FM4Science — ICLR 2026 Workshop FM4Science Poster_

### Official Review · Reviewer_2UGE · 2026-02-24
**Promising direction but with limited generalization**

**Rating:** 7
**Confidence:** 3

**Review:**

The paper proposes CLIBD-HiR and CLIBD-HiR-Fuse, two hierarchy-aware multimodal frameworks for taxonomic inference from specimen images, DNA barcodes, and taxonomic text. The core contribution is a Hierarchical Information Regularization (HiR) loss that enforces consistency across taxonomic levels in the embedding space, combined with a lightweight fusion module for handling noisy and missing modalities. Experimental results on BIOSCAN-1M indicate improved robustness and classification accuracy over CLIBD and simple fusion baselines under synthetic corruption.
Strengths:
1. Addresses an important limitation of existing multimodal biodiversity models by explicitly modeling taxonomic hierarchy.
2. Proposes a principled hierarchical contrastive regularization mechanism grounded in prior supervised contrastive learning work.
3. Demonstrates consistent robustness gains under DNA and image degradation.
4. Evaluates the approach at multiple taxonomic levels (order, family, genus, species).
5. Uses large-scale data and strong pretrained backbones (BioCLIP, DNABERT2).

Weaknesses:
1. The evaluation relies on a closed-set split in which all taxa appearing in the test set are also seen during training. This deviates from the original CLIBD seen/unseen protocol and does not reflect realistic biodiversity scenarios characterized by long-tailed distributions and frequent appearance of novel species. As a result, the experiments mainly assess performance on known classes and provide limited evidence for true generalization.
2. The hierarchical regularization is applied only to image embeddings, while DNA and text representations are excluded from hierarchy-aware supervision without sufficient justification.
3. The proposed HiR loss is closely related to existing hierarchical and supervised contrastive learning methods. The conceptual novelty beyond adapting these techniques to CLIBD is limited.
4. The fusion module yields only modest improvements over simple embedding averaging in most experimental conditions, making the tradeoff between additional architectural complexity and performance gains unclear.
5. Multiple hyperparameters are introduced without sensitivity analysis, making it difficult to assess robustness and reproducibility.
6. Claims related to “foundation models” and real-world deployment are not fully supported by the restricted experimental setting.

---

### Official Review · Reviewer_jc1g · 2026-02-24
**Conceptually Strong Biological Grounding, but lacks Generalisation Evaluation**

**Rating:** 6
**Confidence:** 3

**Review:**

**Summary of the Paper**

The authors propose a multimodal framework (CLIBD-HiR and CLIBD-HiR-Fuse) to improve taxonomic inference from specimen images, DNA barcodes, and text. To overcome the limitations of treating taxonomy as a flat label space, they introduce Hierarchical Information Regularization (HiR), which constrains the contrastive embedding geometry to respect biological taxonomy (order, family, genus, species). Additionally, they train a gated fusion predictor to dynamically handle missing or corrupted modalities. The models are evaluated on a modified, closed-set split of the BIOSCAN-IM dataset, where they demonstrate substantial performance gains over the CLIBD baseline, particularly under noise degradation

**Pros**

•  **Strong Integration of Domain Knowledge**: The Hierarchical Information Regularization (HiR) loss successfully translates biological inductive biases into a mathematical constraint. By enforcing that fine-grained distinctions do not collapse higher-level taxonomic neighborhoods, the model creates an intrinsically noise-stabilized geometry where semantic drift is bounded by coarser taxonomic levels

•  **Practical Robustness for Scientific Deployment**: The paper directly tackles the messy reality of operational biology pipelines, explicitly modeling artifacts like ambiguous DNA bases, partial sequencing reads, and visually cluttered field imagery. The adaptive fusion module elegantly handles these variable modality qualities, supporting flexible image-only, DNA-only, and joint inferences.

**Cons**

•  **Closed-Set Evaluation Limits**: The current experimental split is strictly closed-set, meaning every order, family, genus, and species in the test set is also present in the training set. For a foundation model aimed at global biodiversity, evaluating zero-shot transfer or open-set performance on unseen taxa is critical. The authors acknowledge this limitation regarding the original CLIBD split's seen/unseen structure, but it remains a gap in assessing generalization.

•  **Unexplored Hierarchy Weighting**: The hierarchical loss aggregates rectified pair losses across levels using non-negative weights, but uniform weights are utilized in the experiments. Exploring biologically informed weighting schemes or learning these parameters could potentially yield an even more accurate geometric representation of evolutionary distance.

---

### Meta-Review · Area_Chair_jQbL · 2026-02-28

**Recommendation:** Accept (Poster)
**Confidence:** 4

**Metareview:**

This paper proposes hierarchy-aware multimodal learning for biodiversity taxonomic inference. The Hierarchical Information Regularization (HiR) loss effectively encodes biological taxonomy into embedding geometry, yielding improved robustness under noise and modality corruption. Reviewers find the biological grounding strong and the robustness experiments convincing.

However, evaluation is conducted under a closed-set split in which all taxa are seen during training, limiting conclusions about generalization to unseen species. The conceptual novelty is moderate, and the fusion module provides modest gains over simpler baselines.

---

### Decision · Program_Chairs · 2026-03-03

Accept (Poster)